# FlashOptim: Optimizers for Memory-Efficient Training

**Jose Javier Gonzalez Ortiz** [* 1]  **Abhay Gupta** [1]  **Christopher Rinard** [1 2]  **Davis Blalock** [* 1 3]

## Abstract

Standard mixed-precision training of neural networks requires many bytes of accelerator memory for each model parameter. These bytes reflect not just the parameter itself, but also its gradient and one or more optimizer state variables. With each of these values typically requiring 4 bytes, training even a 7 billion parameter model can be impractical for researchers with less than 100 GiB of accelerator memory. We introduce FlashOptim, a suite of optimizations that reduces per-parameter memory by over 50% while preserving model quality and API compatibility. Our approach introduces two key techniques. First, we improve master weight splitting by finding and exploiting a tight bound on its quantization error. Second, we design companding functions that greatly reduce the error in 8-bit optimizer state quantization. Together with 16-bit gradients, these techniques reduce AdamW memory from 16 bytes to 7 bytes per parameter, or 5 bytes with gradient release. They also cut model checkpoint sizes by more than half. Experiments with FlashOptim applied to SGD, AdamW, and Lion show no measurable quality degradation across a collection of standard vision and language benchmarks, including Llama-3.1-8B finetuning. Our implementation is available at https://github.com/databricks/flashoptim.

## 1. Introduction

Recent advances in deep learning have been driven largely by scaling: larger models trained on more data consistently yield better results across language (Kaplan et al., 2020; Hoffmann et al., 2022; Chowdhery et al., 2023) and vision (Rosenfeld et al., 2020; Tan & Le, 2019; Dehghani et al., 2023) domains. Training large models can require

Table 1. **Memory per parameter (bytes) for model training**. FlashOptim reduces Adam from 16 to 7 bytes and SGD from 12 to 6 bytes. ($\star$) With gradient release, we further reduce total memory requirements by 2 bytes.

| Tensor | SGD | FlashSGD | Adam | FlashAdam |
|---|---|---|---|---|
| Master Weights | 4 | 2 | 4 | 2 |
| Weight Correction | | 1 | | 1 |
| Gradients | 4 | 2 (0$^\star$) | 4 | 2 (0$^\star$) |
| Momentum | 4 | 1 | 4 | 1 |
| Variance | | | 4 | 1 |
| **Total** | **12** | **6 (4$^\star$)** | **16** | **7 (5$^\star$)** |

a great deal of accelerator memory, with each training iteration requiring memory to store parameters, activations, gradients, and optimizer state.

How much memory do these tensors require? Table 1 shows a typical breakdown. Excluding activations, which scale with batch size rather than parameter count, training with Adam uses about 16 bytes per parameter. Training a 7-billion-parameter LLM therefore requires at least 112 GiB of accelerator memory, plus more memory for activations.

Several approaches mitigate this memory consumption. Distributed training with tensor sharding (Rajbhandari et al., 2020) divides the memory load across multiple accelerators. While this is standard practice in well-resourced organizations, it requires access to multiple accelerators that many practitioners lack. A second approach is CPU offloading (Ren et al., 2021), which moves some tensors to host memory at the cost of added overhead and complexity. Third, parameter-efficient methods (Li & Liang, 2021; Hu et al., 2022) reduce trainable parameters by freezing most weights and training either a small subset of the original weights or a small set of new auxiliary weights, but fundamentally alter the training dynamics (Biderman et al., 2024).

In this work, we describe FlashOptim, a set of techniques to reduce parameter-associated memory in common deep learning optimizers. Figure 1 shows an example: with FlashOptim, finetuning Llama-3.1-8B drops from 175 GiB to 113 GiB peak memory. Crucially, these memory savings are effectively free: FlashOptim runs just as fast as standard

---

[*]Equal contribution [1]Databricks AI Research [2]Now at Standard Kernel Co. [3]Now at Google DeepMind. Correspondence to: Davis Blalock <daviswblalock@gmail.com>.

*Proceedings of the 43$^{rd}$ International Conference on Machine Learning*, Seoul, South Korea. PMLR 306, 2026. Copyright 2026 by the author(s).

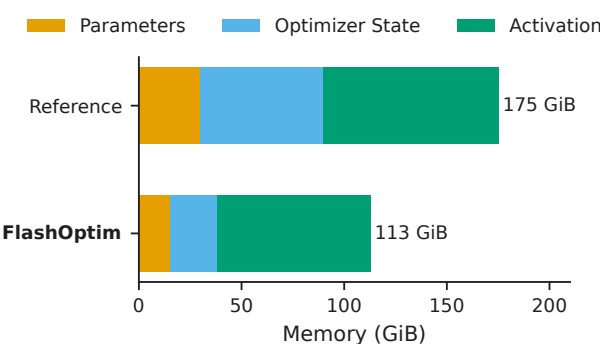

*Figure 1.* **Memory breakdown for finetuning Llama-3.1-8B.** FlashOptim reduces peak memory from 175 to 113 GiB by compressing parameters and optimizer states.

optimizers and causes no measurable loss of model quality across a suite of established training tasks (§4). This allows our optimizer implementations to serve as drop-in replacements for their unoptimized counterparts. FlashOptim incorporates existing enhancements, such as gradient release (Zhang et al., 2023; Warner, 2024), while also introducing improved float splitting (Zamirai et al., 2020; Warner, 2024) and simplified 8-bit optimizer state quantization (Dettmers et al., 2022; Peng et al., 2023; Xi et al., 2025; Fishman et al., 2025). Furthermore, FlashOptim composes cleanly with existing memory-reduction techniques, such as sharding tensors across accelerators, offloading to CPU, or freezing parameters.

We make the following contributions:

- **Improved float splitting**: Instead of materializing both a 32-bit master weight and a 16-bit downcast weight for forward and backward, one can split each master weight into a low-precision weight and a correction term stored in the optimizer (Zamirai et al., 2020; Warner, 2024). We improve on existing float splitting techniques by (a) enabling either 8- or 16-bit error correction and (b) achieving much lower reconstruction error for a given number of correction bits. This allows us to use 24-bit master weights with no loss of model quality.

- **Companded optimizer state quantization**: Several works have shown that one can compress optimizer states to 8 bits per element given sufficient software complexity. We demonstrate that one can do this much more simply, with nothing more than a one-line preprocessing function before standard group-wise linear quantization. Our ablations across different tensor types suggest that designing custom companding functions is a fruitful direction for future research.

- **Fused optimized kernels**: We implement FlashOptim as optimizer step kernels that fuse all compression and quantization operations, reducing memory while

preserving throughput during training. Our implementation is publicly available at `https://github.com/databricks/flashoptim`.

**Conflict of Interest Disclosure.** The authors declare no conflicts of interest.

## 2. Related Work

**Low-Precision Training.** Mixed-precision training (Micikevicius et al., 2018) executes forward and backward passes in FP16 to reduce memory and compute, while retaining FP32 precision for optimizer states and master weights to preserve numerical stability. Kalamkar et al. (2019) showed that BFloat16 (Google, 2019) works equally well, and Zamirai et al. (2020) explored pure BF16 master weights with stochastic rounding and Kahan summation. Recent work has pushed further with FP8 training (Wang et al., 2018; Mellempudi et al., 2019; Micikevicius et al., 2022; Fishman et al., 2025; Narayan et al., 2025), though these approaches primarily target compute formats and retain higher-precision storage for master weights. FlashOptim extends this line of work with an improved float splitting mechanism that reduces storage to 3 bytes per parameter, down from 4-byte FP32, while maintaining FP32-equivalent training semantics.

**Optimizer State Compression.** Dettmers et al. (2022) applied 8-bit block-wise dynamic quantization to Adam's momentum and variance, reducing optimizer state from 8 to 2 bytes per parameter. Follow-up work explored FP8 representations (Peng et al., 2023; Xi et al., 2025; Fishman et al., 2025), and Li et al. (2023) compressed both moments to 4 bits using row and column-wise quantization. MicroAdam (Modoranu et al., 2024) instead compresses gradients before updating optimizer states. Rather than design elaborate quantization methods or number formats, we show that one can obtain 8-bit optimizer states with no quality loss using simple, one-line preprocessing functions. Beyond optimizer states, we address additional sources of per-parameter memory, eliminating entire bytes from other tensors.

**Gradient Memory and Communication.** LOMO (Lv et al., 2024b), AdaLOMO (Lv et al., 2024a), and Adam Accumulation (Zhang et al., 2023) fuse parameter updates into the backward pass to release gradient memory eagerly. However, this conflicts with gradient accumulation, which requires the full accumulated gradient before updating. In distributed settings, gradient communication can also become a bottleneck. One can reduce this bottleneck by, e.g., compressing gradients to 1-bit with error feedback (Tang et al., 2021), or using low-rank approximations (Vogels et al., 2019). FlashOptim supports gradient release when compatible and could be used alongside communication

compression techniques.

**Memory-Efficient Optimization.** Alternative optimizer designs reduce memory by restructuring update rules and stored buffers. Adafactor (Shazeer & Stern, 2018) achieves sublinear memory by factorizing the second moment into row and column statistics; SM3 (Anil et al., 2019) stores structured maxima; NovoGrad (Ginsburg et al., 2019) replaces per-parameter variance with layer-wise normalization. Adam-mini (Zhang et al., 2025) shares variance terms across parameter blocks, while Adapprox (Zhao et al., 2024b) uses a low-rank approximation. Other approaches eliminate the second moment entirely: Lion (Chen et al., 2023) uses sign-based momentum, and Muon (Jordan et al., 2024; Liu et al., 2025) applies orthogonalized updates. Pethick et al. (2025) extend Muon to unify gradient accumulation with momentum, removing dedicated optimizer memory altogether.

Low-rank decompositions approximate full tensors while requiring less memory. For fine-tuning, LoRA (Hu et al., 2022) and QLoRA (Dettmers et al., 2023) freeze base weights and train only low-rank adapters. For pretraining, GaLore (Zhao et al., 2024a) projects gradients to a low-rank subspace, and APOLLO (Zhu et al., 2025) approximates adaptive scaling with random projections. Unlike these approaches that modify the optimizer's update rule, FlashOptim preserves standard optimizer semantics and can be combined with these techniques.

**System-Level Memory Optimizations.** System-level approaches reduce accelerator memory without changing optimization semantics. Activation checkpointing (Chen et al., 2016; Korthikanti et al., 2023) trades compute for memory by recomputing activations during the backward pass. ZeRO (Rajbhandari et al., 2020) partitions optimizer states, gradients, and parameters across data-parallel ranks, while offloading (Rajbhandari et al., 2021; Ren et al., 2021) moves state to CPU or NVMe memory. FlashOptim is orthogonal to these approaches: it reduces the per-rank footprint and can be used with ZeRO, FSDP (Zhao et al., 2023), and activation checkpointing.

## 3. Method

This section describes the two key techniques behind FlashOptim: weight splitting (§3.1) and companded optimizer state quantization (§3.2). We then describe how to integrate these ideas into common optimizer updates while minimizing associated overhead (§3.3).

### 3.1. Weight Splitting

Mixed-precision training uses 16-bit weights for forward and backward passes, but accumulating gradient updates requires higher precision to avoid stagnation (Micikevicius

et al., 2018). Thus, the standard practice is to maintain FP32 precision master weights during training.

However, this introduces waste: the downcast weights take space, but store no information beyond what is saved in the master weights. To eliminate this redundancy, weight splitting (Zamirai et al., 2020; Warner, 2024) instead stores the downcast weights and narrow error-correction values. By combining a 16-bit weight $\theta'$ with a 16-bit error correction value $\rho$, one has enough information to reconstruct a 32-bit master weight $\theta$ with no redundancy.

The core questions in a weight splitting scheme are 1) how to set $(\theta', \rho)$ given $\theta$ and 2) how to estimate $\theta$ given $(\theta', \rho)$. One obvious approach is to use the high 16 bits of $\theta$ as $\theta'$ and the low 16 bits as $\rho$. This admits exact reconstruction of $\theta$ for the special case of BF16 $\theta'$ and FP32 $\theta$, since these formats happen to share the same exponent sizes and offsets. However, this approach does not generalize to other pairs of number formats. It also rounds towards zero instead of towards the nearest low-precision value.

A more general alternative, used in previous work (Zamirai et al., 2020; Warner, 2024), is to set $\rho = \theta - \theta'$, represented as a BF16 value. However, the difference of two floating-point numbers requires as many bits as the wider of the two floats to store exactly,[1] so a BF16 $\rho$ incurs approximation error. For example, if the rounding error were 1e-5, BF16 could only represent 1.0014e-5. In general, while BF16's wide exponent lets it represent nearly the full range of FP32, its 7 mantissa bits only guarantee a relative error bound of 1/256.

Our observation is that **all exponent bits in this scheme are wasted**. The exponent of $e \triangleq \theta - \theta'$ can always be inferred from $\theta'$: under round-to-nearest, $\theta$ must lie within $[\theta' - u/2, \theta' + u/2]$, where $u = \text{ULP}(\theta')$ is the unit in the last place (Goldberg, 1991). If $\theta$ were outside this interval, it would have rounded to a different value. It therefore suffices to encode where $e$ falls within this tiny interval rather than across the full FP32 range.

To exploit this observation, we rescale $e$ such that $[-u/2, u/2]$ maps to $[-N, N]$, where $N \triangleq 2^{b-1} - 1$ for a signed $b$-bit correction integer, and then quantize this rescaled $e$ to the nearest signed $b$-bit integer. That is,

$$\theta' = \text{downcast}(\theta)$$
$$\rho = \text{round}\left(\frac{\theta - \theta'}{\text{ULP}(\theta')/2} \cdot N\right), \quad (1)$$

To reconstruct $\theta$ from $(\theta', \rho)$, we invert this scaling and add

---

[1]Consider, e.g., storing the minimum float32 subnormal, which will be rounded to zero by any narrower datatype.

the result to $\theta'$.

$$\hat{\theta} = \theta' + \frac{\rho}{N} \cdot \frac{\text{ULP}(\theta')}{2} \qquad (2)$$

For tensors of floating-point values, we apply this transformation elementwise. Algorithm 1 provides a lower-level description of the compression and decompression operations with numerical precision considerations.

---

**Algorithm 1** Weight Splitting

---

**Constants:** $N = 2^{b-1} - 1$ (127 for INT8, 32767 for INT16)

1: **function** $\mathcal{C}(\theta)$
2:      $\theta' \leftarrow \text{Downcast}(\theta)$
3:      $e \leftarrow \theta - \text{Float32}(\theta')$
4:      $\ell \leftarrow \lfloor \log_2 \text{ULP}(\theta') \rfloor - 1$
5:      $h \leftarrow \lfloor -\ell/2 \rfloor$ // For numerical stability
6:      $e_{\text{norm}} \leftarrow (e \cdot 2^h) \cdot 2^{-\ell-h}$
7:      $\rho \leftarrow \text{Int}(\text{Round}(\text{Clamp}(e_{\text{norm}}, -1, 1) \cdot N))$
8:      **return** $\theta', \rho$

9: **function** $\mathcal{C}^{-1}(\theta', \rho)$
10:      $\ell \leftarrow \lfloor \log_2 \text{ULP}(\theta') \rfloor - 1$
11:      $h \leftarrow \lfloor \ell/2 \rfloor$
12:      $e \leftarrow ((\text{Float32}(\rho)/N) \cdot 2^h) \cdot 2^{\ell-h}$
13:      **return** $\text{Float32}(\theta') + e$

---

When using BF16 for $\theta'$ and INT8 for $\rho$, the compressed representation provides approximately 24 bits of effective precision (16 from BF16 plus 8 from the error term). Operationally, the model stores BF16 parameters and the optimizer stores the INT8 correction term; fused optimizer kernels reconstruct $\theta$ only inside the update and then write back the compressed pair. This is analogous to the PXR24 format used in high-dynamic-range imaging, which achieves similar precision by rounding 32-bit floats to 24 bits (Kainz et al., 2004).

### 3.2. Companded Optimizer State Quantization

For optimizer state variables such as momentum and variance estimates, a common approach is group-wise quantization: dividing tensors into fixed-length groups and mapping values to a lower-precision format like INT8 (Dettmers et al., 2022). To increase the precision range of the group of values, they are rescaled using the maximum absolute value (*absmax*), which is stored as an additional scale with 32 or 16 bits of precision. While simple, this *uniform* quantization allocates bins evenly across the value range, implicitly assuming that values are roughly uniformly distributed. Our measurements show that optimizer state distributions violate this assumption (§4.5), and we find that applying nonlinear *companding* functions before quantization can reshape these distributions toward uniformity, improving utilization of quantization bins and reducing quantization error. As

---

**Algorithm 2** $\mathcal{Q}_m$: Momentum Quantization

---

**Constants:** $G = 32$, $\epsilon_s = 10^{-12}$

1: **function** $\mathcal{Q}_m(m)$
2:      **for** each group $g$ of $G$ elements **do**
3:          $s_g \leftarrow \max(\max(|m_g|), \epsilon_s)$
4:          $m'_g \leftarrow m_g/s_g$
5:          $m''_g \leftarrow 2m'_g/(1 + |m'_g|)$
6:          $m^q_g \leftarrow \text{Round}(m''_g \cdot 127, \text{INT8})$
7:      **return** $m^q, s$

8: **function** $\mathcal{Q}_m^{-1}(m^q, s)$
9:      **for** each group $g$ **do**
10:         $m''_g \leftarrow m^q_g/127$
11:         $m'_g \leftarrow m''_g/(2 - |m''_g|)$
12:         $m_g \leftarrow m'_g \cdot s_g$
13:      **return** $m$

---

we show in §4.5, this companding step is critical: without it, linear quantization of optimizer states causes training to diverge.

We design specialized transformations for each optimizer state type. The optimal choice of companding function is distribution-dependent, so we selected these functions empirically for the optimizer-state tensors we observed, subject to the constraint that they remain cheap and invertible inside fused kernels. For momentum tensors (used in SGD, Adam, and Lion), we first normalize each group by its absmax scale, then apply a softsign-like function:

$$\phi_m(x) = \frac{2x}{1 + |x|} \qquad \phi_m^{-1}(z) = \frac{z}{2 - |z|} \qquad (3)$$

This function expands small values around zero while leaving the endpoints fixed. This is motivated by the fact that momentum values tend to concentrate near zero, so the transformation spreads the distribution more evenly across quantization bins. In contrast, for variance tensors in Adam, we first apply a square root, then normalize by the group absmax:

$$\phi_v(x) = \sqrt{x} \qquad \phi_v^{-1}(z) = z^2 \qquad (4)$$

Here the square root is motivated by Adam's variance update $v_t = \beta_2 v_{t-1} + (1 - \beta_2)g^2$, which accumulates squared gradients, producing heavy-tailed distributions with large dynamic range. Both transformations satisfy key design criteria: they are exactly invertible, computationally efficient in both directions (one division or square root per element), and require no hyperparameters.

For both momentum and variance, we partition the tensor into groups of $G = 32$ elements and store a separate FP16 scale factor per group, introducing an overhead of $2/G = 1/16$ bytes per parameter. We store the normalized momentum in signed integers (INT8) and variance in unsigned integers (UINT8) since variance is non-negative.

**Algorithm 3** $\mathcal{Q}_v$: Variance Quantization

> **Constants:** $G = 32$, $\epsilon_s = 10^{-12}$

1: **function** $\mathcal{Q}_v(v)$
2:    $v' \leftarrow \sqrt{v}$
3:    **for** each group $g$ of $G$ elements **do**
4:       $s_g \leftarrow \max(\max(v'_g), \epsilon_s)$
5:       $v^q_g \leftarrow \text{Round}((v'_g/s_g) \cdot 255, \text{UINT8})$
6:    **return** $v^q, s$

7: **function** $\mathcal{Q}_v^{-1}(v^q, s)$
8:    **for** each group $g$ **do**
9:       $v'_g \leftarrow (v^q_g/255) \cdot s_g$
10:      $v_g \leftarrow (v'_g)^2$
11:    **return** $v$

Algorithms 2 and 3 detail the quantization and dequantization procedures for momentum and variance, respectively.

### 3.3. Optimizer Update

We modify any given gradient update rule by adding a prologue and an epilogue. In the prologue, we dequantize the optimizer states and reconstruct the master weight $\theta$ from the low-precision weight $\theta'$ and the error correction bits $\rho$. In the epilogue, we quantize the new optimizer state and split the new $\theta$ into an updated $(\theta', \rho)$. At the start of training, we downcast the master weights to BF16 to ensure training runs directly on the low-precision $\theta'$ with no downcasts apart from our optimizer step. Algorithm 4 illustrates these changes for the AdamW optimizer, and Algorithms 5 and 6 in the appendix show the corresponding changes to SGD and Lion respectively.

### 3.4. Implementation

**Update kernels.** Since compression and quantization are bandwidth-bound operations, we implement the optimizer step as a single fused Triton kernel (Tillet et al., 2019). For example, for the AdamW update, our kernel encompasses steps 9 through 22 from Algorithm 4.

**Gradient release.** We implement gradient release (Zhang et al., 2023), interleaving gradient computation with optimizer updates during backpropagation. As each gradient is computed, we eagerly apply the optimizer rule to free the gradient memory. We apply this optimization only when gradient accumulation is disabled.

**Distributed training.** Our implementation is compatible with parameter sharding approaches such as PyTorch FSDP (Zhao et al., 2023). During forward and backward passes, only the 16-bit $\theta'$ parameters are all-gathered; the correction term $\rho$ remains local with the optimizer states.

**Checkpoint size.** Our representation reduces checkpoint

**Algorithm 4** FlashAdamW: Memory-Efficient AdamW. We highlight the changes from AdamW.

**Require:** Parameters $\theta_0$, learning rate schedule $\{\eta_t\}_{t=1}^T$, $\beta_1, \beta_2 \in [0, 1)$, $\varepsilon > 0$, weight decay $\lambda \geq 0$, loss $\mathcal{L}(\theta)$, minibatch sampler $\mathcal{B}(\cdot)$

1:  $m^q_0, m^s_0 \leftarrow \mathcal{Q}_m(0)$
2:  $v^q_0, v^s_0 \leftarrow \mathcal{Q}_v(0)$
3:  $\theta'_0, \rho_0 \leftarrow \mathcal{C}(\theta_0)$
4: Initialize $t \leftarrow 0$
5: **while** not converged **do**
6:    $t \leftarrow t + 1$
7:    $B_t \sim \mathcal{B}$
8:    $g_t \leftarrow \nabla_\theta \mathcal{L}(B_t; \theta'_{t-1})$
9:    // Reconstruct optimizer state and master weight
10:   $m_{t-1} \leftarrow \mathcal{Q}_m^{-1}(m^q_{t-1}, m^s_{t-1})$
11:   $v_{t-1} \leftarrow \mathcal{Q}_v^{-1}(v^q_{t-1}, v^s_{t-1})$
12:   $\theta_{t-1} \leftarrow \mathcal{C}^{-1}(\theta'_{t-1}, \rho_{t-1})$
13:   // Standard optimizer update
14:   $m_t \leftarrow \beta_1 m_{t-1} + (1 - \beta_1)g_t$
15:   $v_t \leftarrow \beta_2 v_{t-1} + (1 - \beta_2)g_t^2$
16:   $\hat{m}_t \leftarrow m_t/(1 - \beta_1^t)$
17:   $\hat{v}_t \leftarrow v_t/(1 - \beta_2^t)$
18:   $\theta_t \leftarrow \theta_{t-1} - \eta_t \left(\hat{m}_t/(\sqrt{\hat{v}_t} + \varepsilon) + \lambda\theta_{t-1}\right)$
19:   // Quantize optimizer state and split master weight
20:   $m^q_t, m^s_t \leftarrow \mathcal{Q}_m(m_t)$
21:   $v^q_t, v^s_t \leftarrow \mathcal{Q}_v(v_t)$
22:   $\theta'_t, \rho_t \leftarrow \mathcal{C}(\theta_t)$

size. For instance, standard Adam checkpoints require 12 bytes per parameter (4 for weights, 4 for momentum, 4 for variance); FlashAdamW requires only 5 bytes (2 for weights, 1 for correction, 1 for momentum, and 1 for variance). For a 7B model, checkpoint size reduces from 84 GiB to 35 GiB.

**Code availability.** We make our implementation widely available as an open-source PyTorch library at `https://github.com/databricks/flashoptim`.

## 4. Experiments

### 4.1. Experimental Setup

We evaluate FlashOptim with three optimizers: SGD with momentum (Polyak, 1964), AdamW (Loshchilov & Hutter, 2019), and Lion (Chen et al., 2023). We refer to these variants as FlashSGD, FlashAdamW, and FlashLion. We test these across image classification, language model pretraining, and supervised finetuning tasks.

To ensure a fair comparison, all experiments use identical hyperparameters between reference optimizers and their FlashOptim counterparts. We re-implement the reference optimizers with similar fused Triton kernels for consistent

measurement of memory and throughput, and all reference implementations use mixed precision (Micikevicius et al., 2018) to keep activations in 16-bit precision. The precisions of master weights $\theta$, error correction term $\rho$, gradients $g$, momentum $m$, variance $v$, and activations $a$ are as follows:

|            | $\theta$ | $\rho$ | $g$  | $m$  | $v$   | $a$  |
|------------|----------|--------|------|------|-------|------|
| Reference  | FP32     | —      | FP32 | FP32 | FP32  | BF16 |
| FlashOptim | BF16     | INT8   | BF16 | INT8 | UINT8 | BF16 |

Our experiments demonstrate four main findings. First, FlashOptim matches reference convergence and accuracy across all tested configurations (§4.2). Second, it reduces optimizer memory by over 50% with negligible computational overhead (§4.3). Third, our ULP-based weight splitting achieves near-optimal reconstruction (§4.4). Finally, our companding functions significantly reduce quantization error for both momentum and variance states (§4.5).

**Image Classification.** We train a ResNet-50 architecture (He et al., 2016b) on the ILSVRC2012 (ImageNet-1K) dataset (Deng et al., 2009). We use the hyperparameters recommended by Nvidia (NVIDIA, 2023), with additional details provided in Appendix B.1.

**LLM Pretraining.** We evaluate LLM pretraining using the training recipe outlined in the nanoGPT repository (Karpathy, 2023). We use the GPT-2 (Radford et al., 2019) architecture and train on a 10B token subset of the FineWeb dataset (Penedo et al., 2024), following the setup of Jordan (2024). We provide hyperparameter details in Appendix B.2. We evaluate models using a suite of in-context learning (ICL) benchmarks that assess commonsense reasoning and language understanding capabilities. We provide a complete list of benchmarks in Appendix B.3.

**LLM Finetuning.** We run supervised finetuning on a pretrained Llama-3.1-8B model (Dubey et al., 2024) on OpenMathInstruct-2 (Toshniwal et al., 2024), and evaluate on the GSM8k (Cobbe et al., 2021) benchmark. We provide further hyperparameter details in Appendix B.4.

**Compressed Baselines.** For the AdamW language experiments, we also compare against Dettmers 2022 (Dettmers et al., 2022), which quantizes optimizer state to 8-bit integer tensors while retaining FP32 master weights, and Zamirai 2020 (Zamirai et al., 2020; Warner, 2024), a technique that splits the parameters into BF16 master weights and BF16 error correction terms based on Kahan summation. When reporting these methods, we use the same training and evaluation setup as the corresponding AdamW and FlashAdamW runs.

**Training and Infrastructure.** We train with distributed data parallelism for the image classification and LLM pretraining tasks, and for LLM finetuning we use FSDP (Zhao et al., 2023) and activation checkpointing. We train all mod-

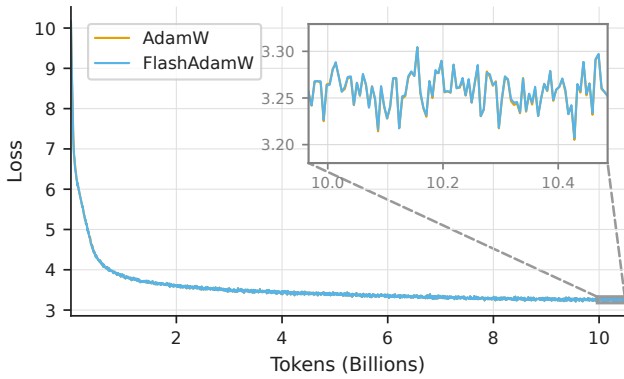

*(a)* LLM Pretraining (GPT-2 + AdamW)

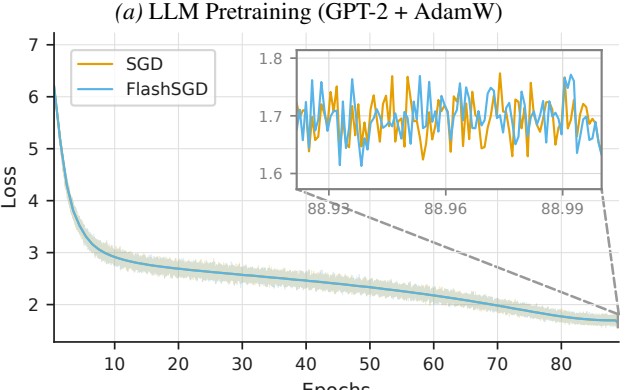

*(b)* Image Classification (ResNet-50 + SGD)

*Figure 2.* **Training convergence**. Comparison of training loss trajectories between reference optimizers and their FlashOptim variants. Both achieve nearly identical loss curves throughout training, demonstrating that our memory optimizations do not impact model quality.

els using PyTorch 2.8 and CUDA 12.8 on NVIDIA H100 GPUs. We report the mean and standard deviation for all our results with 3 random seeds. Within each seed, all compared variants start from identical weights and consume the same minibatches in the same order. System metrics (memory, timing) are measured in steady-state.

### 4.2. Convergence and Accuracy

We first verify that FlashOptim introduces no measurable degradation by comparing training convergence and validation accuracy. Figure 2a shows training loss for LLM pretraining with AdamW and FlashAdamW. FlashAdamW produces a nearly identical trajectory to the reference AdamW and closely tracks AdamW even after 20,000 parameter updates, indicating that reduced precision does not affect learning dynamics. Figure 2b shows similar results for image classification: FlashSGD matches reference SGD throughout training. For LLM finetuning, Figure 10 in the appendix shows an analogous result for AdamW. Appendix B.2 evaluates the same AdamW comparison at larger pretraining scales: GPT-2 Large (~0.8B parameters) and a LLaMA-

*Table 2.* **Image Classification Results**. Validation accuracy for ResNet-50. We report standard deviation across 3 training runs. FlashOptim matches the reference scores for both SGD and AdamW.

| | ImageNet Top-1 Acc. | |
| --- | --- | --- |
| | **SGD** | **AdamW** |
| Reference | $77.01 \pm 0.02$ | $75.51 \pm 0.09$ |
| FlashOptim | $77.16 \pm 0.09$ | $75.67 \pm 0.04$ |

*Table 3.* **LLM Finetuning Results**. GSM8k accuracy for Llama-3.1-8B finetuning with AdamW variants. We report standard deviation across 3 training runs. All compressed variants match the reference score.

| Method | GSM8k Acc. |
| --- | --- |
| Reference AdamW | $75.11 \pm 0.66$ |
| FlashOptim | $75.15 \pm 0.49$ |
| Kahan Splitting | $75.06 \pm 0.30$ |
| 8bit-AdamW | $75.06 \pm 0.20$ |

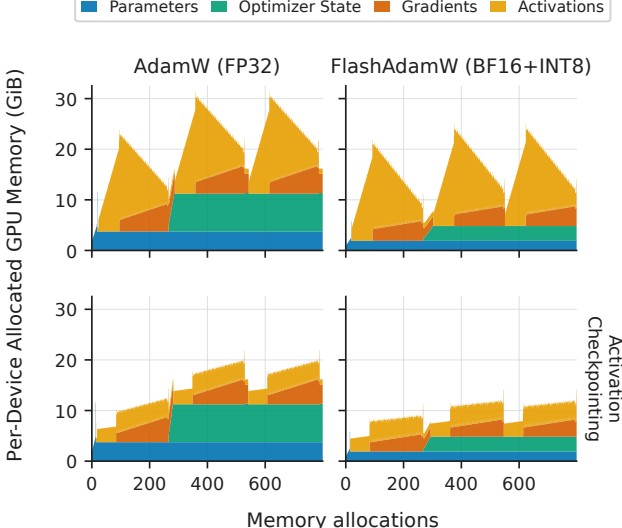

*Figure 3.* **Peak allocated GPU memory over time.** Per-device memory allocation traces over three training steps for Llama-3.1-8B finetuning with FSDP on 8 H100 GPUs, without activation checkpointing (top) and with activation checkpointing (bottom). In both settings, FlashAdamW reduces the persistent parameter and optimizer-state footprint throughout training compared with AdamW.

style decoder-only Transformer ($\sim$1.7B parameters). In the GPT-2 Large runs, FlashAdamW tracks AdamW convergence and preserves ICL quality. In the single LLaMA-style 1.7B run, FlashAdamW closely tracks AdamW training loss and final validation bits-per-byte.

Table 2 reports final scores for image classification, and Table 3 reports GSM8k accuracy for LLM finetuning. FlashSGD and FlashAdamW match the reference optimizer scores in these settings. Table 4 compares final validation loss and in-context learning scores for the LLM pretraining task. Models trained with FlashOptim achieve scores within variance of reference optimizers across all metrics.

### 4.3. Memory and Speed

We compare memory requirements and optimizer step time, demonstrating that FlashOptim reduces peak memory (PyTorch Contributors, 2025) without practical overhead. We focus on parameter-related memory (weights, optimizer state, gradients) since activation memory is identical across both settings. To isolate contributions, we ablate: weight splitting (Weight Split) with full-precision optimizer states, and optimizer state quantization with companding (Opt. Quant.) with FP32 master weights.

Table 5 breaks down the memory usage for LLM finetuning on a Llama-3.1-8B model. As anticipated, we reduce parameter memory by 50% from dropping precision from FP32 to BF16, and reduce optimizer memory by 61% from quantizing the optimizer state tensors. Moreover, when looking at peak memory (including activations), FlashOptim reduces it by 36% with no practical slowdown in optimizer step times. We verified that our reference AdamW implementation is efficient, matching the optimized fused PyTorch AdamW

kernel; for example, on Llama-3.1-8B finetuning, their step times are 10.2 ms and 10.1 ms, respectively.

Ablating each component confirms that weight splitting halves master weight memory while adding 13% of extra optimizer state. Optimizer state quantization reduces optimizer state by $\sim$73%, mapping FP32 tensors to INT8/UINT8; the reduction is slightly less than 75% due to the overhead of storing FP16 scale factors for each group of 32 elements. Table 7 and Table 9 in the appendix include corresponding memory and step-time measurements for image classification and LLM pretraining.

When comparing to existing compression techniques, we find that 8bit-AdamW reduces optimizer memory but keeps FP32 master weights, whereas Kahan Splitting reduces parameter memory but keeps wider optimizer states. FlashOptim is the only compared method that reduces both parameter and optimizer memory while preserving quality and comparable step time.

Figure 3 shows the corresponding peak allocated memory over three training steps, both with and without activation checkpointing, confirming that FlashAdamW lowers the persistent parameter and optimizer-state footprint throughout training. Since reconstruction is fused and blockwise, the temporary FP32 weights are never written to HBM and do not appear as an additional memory spike.

*Table 4.* **LLM Pretraining Results**. NanoGPT results with GPT-2 (124M). We report validation loss and accuracy (%) on in-context learning benchmarks assessing commonsense reasoning and language understanding. We report standard deviation across 3 training runs.

| Optimizer | Val Loss | HellaSwag | ARC-E | CSQA | PIQA | LAMBADA | Winograd | BoolQ | Mean ICL |
|---|---|---|---|---|---|---|---|---|---|
| AdamW | $3.263 \pm 0.001$ | $31.9 \pm 0.2$ | $39.6 \pm 0.7$ | $25.9 \pm 4.1$ | $64.3 \pm 0.0$ | $31.0 \pm 1.2$ | $57.3 \pm 0.6$ | $58.1 \pm 3.6$ | $44.0 \pm 0.4$ |
| FlashAdamW | $3.265 \pm 0.001$ | $31.9 \pm 0.5$ | $39.5 \pm 0.9$ | $30.8 \pm 2.1$ | $64.5 \pm 0.3$ | $31.9 \pm 0.7$ | $59.1 \pm 1.1$ | $57.2 \pm 4.8$ | $45.0 \pm 1.0$ |
| Kahan Splitting | $3.264 \pm 0.001$ | $32.0 \pm 0.2$ | $39.7 \pm 0.4$ | $24.1 \pm 3.7$ | $64.3 \pm 0.8$ | $30.9 \pm 1.0$ | $60.1 \pm 2.5$ | $58.2 \pm 2.7$ | $44.2 \pm 0.5$ |
| 8bit-AdamW | $3.279 \pm 0.000$ | $31.6 \pm 0.3$ | $39.3 \pm 0.9$ | $24.2 \pm 2.7$ | $64.0 \pm 0.5$ | $30.1 \pm 0.6$ | $58.4 \pm 2.4$ | $53.4 \pm 5.6$ | $43.0 \pm 1.6$ |
| Lion | $3.240 \pm 0.002$ | $32.3 \pm 0.0$ | $40.0 \pm 0.5$ | $23.3 \pm 1.8$ | $63.8 \pm 1.0$ | $31.5 \pm 0.2$ | $58.9 \pm 2.0$ | $58.1 \pm 2.4$ | $44.0 \pm 0.5$ |
| FlashLion | $3.240 \pm 0.001$ | $32.4 \pm 0.3$ | $40.8 \pm 0.2$ | $25.4 \pm 2.9$ | $64.2 \pm 0.5$ | $31.6 \pm 0.5$ | $59.1 \pm 2.4$ | $59.1 \pm 2.3$ | $44.7 \pm 0.5$ |

*Table 5.* **Profiling**. Runtime measurements for LLM Finetuning with Llama-3.1-8B. We capture master weight memory (Params), optimizer state memory (Optim), peak GPU memory (Peak), and optimizer step times (Step). We compare FlashOptim to existing compression techniques as well as ablations of our method components.

| | Params | | Optim | | Peak | | Step |
|---|---|---|---|---|---|---|---|
| **Variant** | GiB | Δ | GiB | Δ | GiB | Δ | ms |
| Reference | 29.9 | | 59.8 | | 175.2 | | 10.2 |
| FlashOptim | **15.0** | -50% | **23.4** | -61% | **112.7** | -36% | 9.4 |
| Kahan Splitting | 15.0 | -50% | 44.9 | -25% | 134.2 | -23% | 15.8 |
| 8bit-AdamW | 29.9 | | 15.2 | -75% | 130.6 | -25% | 31.9 |
| *Ablations* | | | | | | | |
| Weight Split | 15.0 | -50% | 67.3 | +13% | 156.7 | -11% | 10.7 |
| Opt. Quant. | 29.9 | | 15.9 | -73% | 131.4 | -25% | 10.5 |

## 4.4. Weight Splitting

We compare our weight splitting scheme to Zamirai et al. (2020), who store the rounding error in a floating-point buffer for Kahan summation error correction. Since both approaches are data-independent, we evaluate them exhaustively over all finite FP32 bitstrings, computing relative error after applying compression and decompression in sequence. We consider four methods: a no error correction baseline, storing error in the same 16-bit format, our ULP-normalized error with 8-bit integers, and ours with 16-bit integers.

Figure 4 plots mean relative error versus exponent for BF16 and FP16. For BF16, our ULP approach with 16-bit correction achieves near-zero error ($< 10^{-9}$). With 16 bits of correction term, we achieve perfect bitwise reconstruction in 99.92% of the values. In contrast, storing the error in BF16 (BF16+BF16) produces substantially worse error ($> 10^{-6}$), comparable to our 24-bit format. For FP16, our 32-bit ULP format perfectly reconstructs values in the normal range and dominates FP16+FP16 throughout. Our 24-bit format produces constant error across the normal range, improving worst-case error from $10^{-4}$ to under $10^{-6}$.

## 4.5. Optimizer State Quantization

We validate our companding functions by comparing quantization error against standard scaled integer quantization. Us-

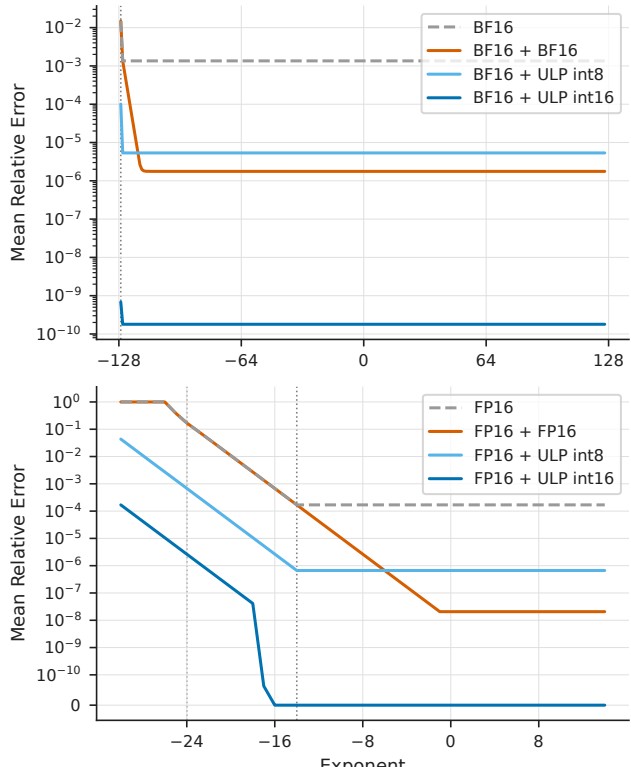

*Figure 4.* **FP32 Reconstruction Error.** Comparison of FP32 reconstruction error for different weight compression schemes across exponent ranges for a target datatype of BF16 (top) and FP16 (bottom). Our ULP-based error correction achieves lower relative error, particularly for small exponents. Denormal floating-point ranges are indicated with vertical dotted lines.

ing a fixed full-precision training trajectory for consistency, we quantize and dequantize momentum and variance buffers at each step, computing normalized MSE (NMSE) against the original values. Figure 5 shows quantile distributions of NMSE for each optimizer and buffer type. Companding reduces error for momentum buffers and provides substantial improvements for variance buffers, where NMSE drops significantly.

Beyond reducing quantization error, in some cases companding is essential for training stability. Figure 6 shows LLM pretraining with and without variance companding: linear

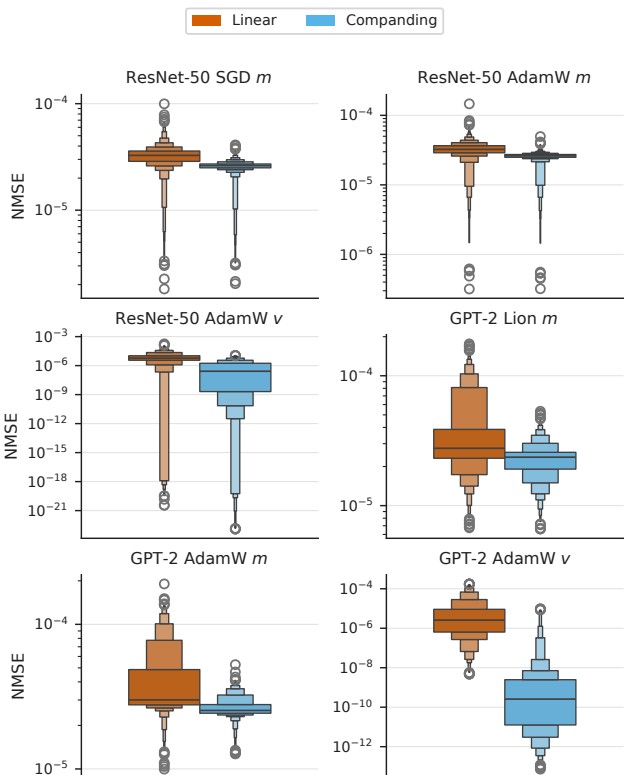

Figure 5. **Optimizer state quantization error.** NMSE comparison between standard scaled integer quantization (Linear) and our companding approach for momentum ($m$) and variance ($v$) buffers across different optimizers and datasets. Companding reduces quantization error across all optimizer types and tensor types, with particularly large improvements for variance tensors.

quantization causes training to diverge, while companding maintains stable convergence.

## 5. Limitations

FlashOptim is designed to minimize parameter-associated memory consumption, so models with large parameter counts and small activations benefit most from our optimizations. Smaller architectures with large activations, such as convolutional networks with high-resolution feature maps, are often dominated by activation memory. In these activation-dominated regimes, a 50% reduction in parameter memory translates to modest total memory savings, and techniques like activation checkpointing are more effective for such workloads.

Another limitation is that some tasks and architectures may be more sensitive to quantization than those in our benchmarks. The effectiveness of a quantization pipeline depends on the data distribution, and there is no guarantee that our method (or any quantization approach) will preserve model quality in all cases. Likely failure regimes include parameters with sparse or intermittent updates, such as rarely used

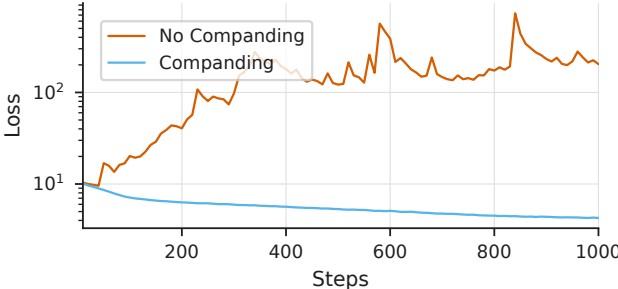

Figure 6. **Companding prevents training divergence.** GPT-2 training with AdamW and quantized optimizer states: linear quantization (no companding) causes rapid divergence, while our companding approach maintains stable training dynamics.

embedding rows, and optimizer states whose distributions differ substantially from those used to design our current companding functions. Consequently, our implementation allows selectively disabling compression or excluding specific layers as needed.

Finally, while we find that 24-bit master weights are sufficient in our experiments, not even 32 bits are guaranteed to suffice in all cases. Successive (normal) floating-point values differ by a factor of roughly $2^{-\text{num\_mantissa\_bits}}$, and if the ratio of weight update to weight magnitude falls below this threshold, the update will be discarded. This risk is most relevant under annealed learning-rate schedules, where the update magnitude relative to the weights being modified can fall below this spacing threshold.

## 6. Conclusion

We introduced FlashOptim, a method to reduce the memory footprint of neural network training while preserving optimizer semantics and model quality. FlashOptim provides drop-in replacements for common optimizers and requires no additional tuning.

Our approach combines two key techniques. First, we reduce master weights from 32 to 24 bits via improved floating-point error correction. Second, we use companding functions to enable 8-bit optimizer state quantization. Together with 16-bit gradients, these reduce per-parameter memory by over 50% for AdamW.

We validated FlashOptim on image and language benchmarks using SGD, AdamW, and Lion. Across the main evaluated settings, our method matches reference implementations in both loss and accuracy while providing significant memory savings. FlashOptim composes with FSDP and activation checkpointing, enabling multiplicative benefits for large-scale training. By lowering memory requirements, FlashOptim enables practitioners and researchers with limited hardware to train larger models than previously feasible.

## Acknowledgements

We are grateful to Jonathan Frankle, Xing Chen, and Matei Zaharia for their continued support and guidance. We also thank Jialu Liu and Erich Elsen for insightful discussions.

## Impact Statement

Our work makes pretraining, midtraining, and finetuning of large models more accessible to those with limited hardware. This helps democratize deep learning research and makes it easier for academics and independent researchers to contribute to the field. The implications of having a branch of science more accessible to smaller players are non-obvious, but we expect they are net positive: there are more eyes to scrutinize claims, more hands to reproduce results, and more brains to invent improvements to the technology. There may also be economic benefits from training more researchers and spreading a useful technology beyond the halls of the best-funded organizations.

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

## A. Method Details

This section provides detailed pseudocode for the FlashOptim version of SGD (Algorithm 5) and Lion (Algorithm 6).

---

**Algorithm 5** FlashSGD: Memory-Efficient SGD. We highlight the changes from SGD.

---

**Require:** Parameters $\theta_0$, learning rate schedule $\{\eta_t\}_{t=1}^T$, momentum $\mu \in [0, 1)$, weight decay $\lambda \geq 0$, loss $\mathcal{L}(\theta)$, minibatch sampler $\mathcal{B}(\cdot)$

1: $m_0^q, m_0^s \leftarrow \mathcal{Q}_m(0)$
2: $\theta_0', \rho_0 \leftarrow \mathcal{C}(\theta_0)$
3: Initialize $t \leftarrow 0$
4: **while** not converged **do**
5:     $t \leftarrow t + 1$
6:     $B_t \sim \mathcal{B}$
7:     $g_t \leftarrow \nabla_\theta \mathcal{L}(B_t; \theta_{t-1}')$
8:     $m_{t-1} \leftarrow \mathcal{Q}_m^{-1}(m_{t-1}^q, m_{t-1}^s)$
9:     $m_t \leftarrow \mu m_{t-1} + g_t$
10:    $m_t^q, m_t^s \leftarrow \mathcal{Q}_m(m_t)$
11:    $\theta_{t-1} \leftarrow \mathcal{C}^{-1}(\theta_{t-1}', \rho_{t-1})$
12:    $\theta_t \leftarrow \theta_{t-1} - \eta_t(m_t + \lambda\theta_{t-1})$
13:    $\theta_t', \rho_t \leftarrow \mathcal{C}(\theta_t)$

---

**Algorithm 6** FlashLion: Memory-Efficient Lion. We highlight the changes from Lion.

---

**Require:** Parameters $\theta_0$, learning rate schedule $\{\eta_t\}_{t=1}^T$, $\beta_1, \beta_2 \in [0, 1)$, weight decay $\lambda \geq 0$, loss $\mathcal{L}(\theta)$, minibatch sampler $\mathcal{B}(\cdot)$

1: $m_0^q, m_0^s \leftarrow \mathcal{Q}_m(0)$
2: $\theta_0', \rho_0 \leftarrow \mathcal{C}(\theta_0)$
3: Initialize $t \leftarrow 0$
4: **while** not converged **do**
5:     $t \leftarrow t + 1$
6:     $B_t \sim \mathcal{B}$
7:     $g_t \leftarrow \nabla_\theta \mathcal{L}(B_t; \theta_{t-1}')$
8:     $m_{t-1} \leftarrow \mathcal{Q}_m^{-1}(m_{t-1}^q, m_{t-1}^s)$
9:     $u_t \leftarrow \text{sign}(\beta_1 m_{t-1} + (1 - \beta_1)g_t)$
10:    $m_t \leftarrow \beta_2 m_{t-1} + (1 - \beta_2)g_t$
11:    $m_t^q, m_t^s \leftarrow \mathcal{Q}_m(m_t)$
12:    $\theta_{t-1} \leftarrow \mathcal{C}^{-1}(\theta_{t-1}', \rho_{t-1})$
13:    $\theta_t \leftarrow \theta_{t-1} - \eta_t(u_t + \lambda\theta_{t-1})$
14:    $\theta_t', \rho_t \leftarrow \mathcal{C}(\theta_t)$

---

## B. Experimental Details

For all our experiments, we use the MosaicML Streaming (MosaicML, 2022) library to ensure deterministic data loading for distributed training.

### B.1. Image Classification

We train the ResNet-50 (He et al., 2016b) model using the timm library on the ILSVRC2012 (ImageNet-1K) dataset (Deng et al., 2009), which contains approximately 1.28 million training images and 50,000 validation images across 1,000 classes. Our implementation of ImageNet follows the standard setup from (Krizhevsky et al., 2012; Simonyan & Zisserman, 2015). The image is resized with its shorter side randomly sampled in $[256, 480]$ for scale augmentation (Simonyan & Zisserman, 2015). A $224 \times 224$ crop is randomly sampled from an image or its horizontal flip, and then normalized. For evaluation, the image is first resized to $256 \times 256$, followed by a $224 \times 224$ center crop, and then normalized. We initialize the network with Kaiming He initialization (He et al., 2016a) and zero-init residuals (He et al., 2016b).

We train for 90 epochs with a batch size of 1024, using a 5-epoch linear warmup followed by cosine learning rate decay, following the recommended settings from (Goyal et al., 2017). We disable weight decay for biases and BatchNorm layers. We apply label smoothing (Szegedy et al., 2016) with coefficient 0.1. Both reference (FP32 master weights) and FlashOptim use BF16 activations. Table 6 summarizes the hyperparameters we use for training.

*Table 6.* Optimizer hyperparameters for ImageNet/ResNet-50.

|  | SGD | AdamW |
|---|---|---|
| Learning Rate | 1.024 | $3 \times 10^{-3}$ |
| Momentum / Betas | 0.9 | (0.9, 0.999) |
| Weight Decay | $3 \times 10^{-5}$ | $3 \times 10^{-4}$ |

**Memory and Speed Profiling.** Table 7 presents the results of the memory and speed profile for training.

*Table 7.* Memory and speed profiling for image classification (ResNet-50).

| Variant | Params GiB | Δ | Optim GiB | Δ | Total GiB | Δ | Step ms |
|---|---|---|---|---|---|---|---|
| **SGD** | | | | | | | |
| Reference | 0.10 | | 0.10 | | 0.30 | | 8.4 |
| FlashOptim | 0.05 | -46% | 0.05 | -45% | 0.17 | -45% | 9.0 |
| Weight Split | 0.05 | -46% | 0.12 | +23% | 0.23 | -22% | 8.7 |
| Opt. Quant. | 0.10 | | 0.03 | -73% | 0.23 | -23% | 8.7 |
| **AdamW** | | | | | | | |
| Reference | 0.10 | | 0.19 | | 0.40 | | 11.9 |
| FlashOptim | 0.05 | -46% | 0.08 | -56% | 0.20 | -50% | 12.2 |
| Weight Split | 0.05 | -46% | 0.21 | +11% | 0.33 | -17% | 12.1 |
| Opt. Quant. | 0.10 | | 0.05 | -73% | 0.25 | -36% | 12.6 |

Figure 7 shows training loss for ResNet-50 with AdamW and FlashAdamW. FlashAdamW produces a nearly identical trajectory to the reference AdamW implementation.

## B.2. LLM Pretraining

We train the GPT-2 (Radford et al., 2019) (124M) architecture with 12 transformer layers, 12 attention heads, embedding dimension 768, and context length of 1024. We train on the FineWeb10B (kjj0, 2024) dataset, a subset of approximately 10 billion tokens from the FineWeb (Penedo et al., 2024) dataset, tokenized using the GPT-2 BPE tokenizer. We train for 20,000 steps with a batch size of roughly 0.5 million tokens per step. We apply learning rate warmup for the first 700 steps, followed by a cosine decay to 0. The global norm is clipped at 1.0 and a weight decay of 0.1 is used. Weight decay is applied only to 2D parameters (i.e., weight matrices and embeddings), excluding biases and layer normalization parameters. We train in BF16 mixed precision. Table 8 summarizes the optimizer hyperparameters.

*Table 8.* Optimizer hyperparameters for GPT-2 124M pretraining.

|  | AdamW | Lion |
|---|---|---|
| Learning Rate | $6 \times 10^{-4}$ | $2 \times 10^{-4}$ |
| Betas | (0.9, 0.95) | (0.9, 0.95) |

**Memory and Speed Profiling.** Table 9 presents the memory and speed profiling results for LLM pretraining.

Figure 8 shows training loss for LLM pretraining with Lion and FlashLion. FlashLion produces a nearly identical trajectory to the reference Lion implementation and closely tracks Lion even after 20,000 parameter updates.

**Larger Pretraining Runs.** We also compare AdamW and FlashAdamW on two larger pretraining settings. Both settings use the FineWeb10B token stream and the same ICL benchmark suite as the GPT-2 124M experiments. For the first setting, we train GPT-2 Large (∼0.8B parameters) from scratch for 10.49B tokens. We run three matched seeds,

*Table 9.* Memory and speed profiling for LLM pretraining (GPT-2 124M).

| Variant | Params GiB | Δ | Optim GiB | Δ | Total GiB | Δ | Step ms |
|---|---|---|---|---|---|---|---|
| **AdamW** | | | | | | | |
| Reference | 0.46 | | 0.93 | | 1.77 | | 5.7 |
| FlashOptim | 0.23 | -50% | 0.36 | -61% | 0.74 | -58% | 5.9 |
| Weight Split | 0.23 | -50% | 1.04 | +12% | 1.43 | -20% | 5.9 |
| Opt. Quant. | 0.46 | | 0.25 | -73% | 1.08 | -39% | 5.8 |
| **Lion** | | | | | | | |
| Reference | 0.46 | | 0.46 | | 1.30 | | 4.3 |
| FlashOptim | 0.23 | -50% | 0.24 | -48% | 0.62 | -53% | 4.5 |
| Weight Split | 0.23 | -50% | 0.58 | +25% | 0.96 | -26% | 4.4 |
| Opt. Quant. | 0.46 | | 0.12 | -73% | 0.96 | -26% | 4.4 |

and each paired seed uses the same initialization and data order. Figure 9 shows seed 18 for readability, while Table 10 reports mean and standard deviation across all three seeds. AdamW and FlashAdamW reach the same final validation loss, 2.981 after rounding to three decimals.

For the second setting, we train a LLaMA-style decoder-only Transformer with ∼1.7B parameters for 10.21B tokens. This model has 26 layers, hidden size 1664, 13 attention heads, rotary embeddings, and sequence length 2048. AdamW and FlashAdamW use identical initialization and data order. AdamW reaches 0.806 final validation bits-per-byte, while FlashAdamW reaches 0.803. We run this setting once, so we report only the convergence and validation-loss comparison rather than single-run ICL scores.

## B.3. In-Context Learning Benchmarks

We evaluate our pretrained language models on a suite of eight in-context learning (Brown et al., 2020) benchmarks that assess diverse commonsense reasoning and language understanding capabilities:

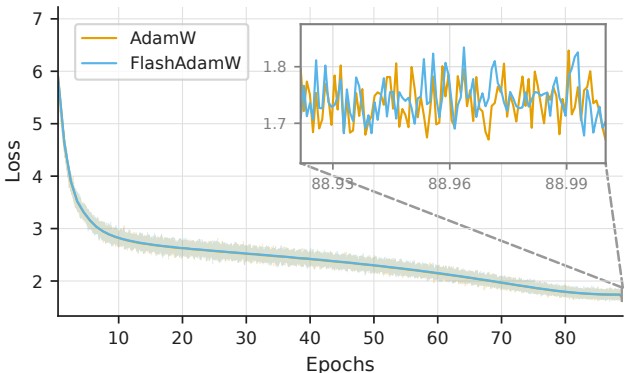

*Figure 7.* **Training convergence for image classification** (ResNet-50 + AdamW). Comparison of training loss between reference AdamW and FlashAdamW on ImageNet.

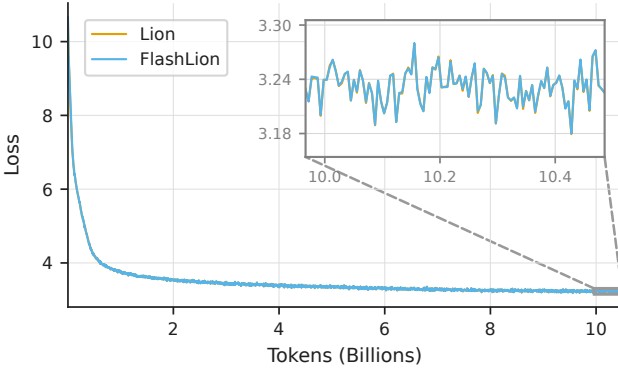

*Figure 8.* **Training convergence for LLM pretraining** (GPT-2 + Lion). Comparison of validation loss between reference Lion and FlashLion on FineWeb10B.

*Table 10.* **Additional GPT-2 Large ICL results.** GPT-2 Large reports mean and standard deviation across three seeds. Scores are zero-shot accuracies (%) on the same benchmark suite used in Table 4.

| Setting | Optimizer | HellaSwag | ARC-E | CSQA | PIQA | LAMBADA | Winograd | BoolQ | Mean ICL |
|---|---|---|---|---|---|---|---|---|---|
| GPT-2 Large | AdamW | 40.0 ± 0.2 | 48.1 ± 0.8 | 22.1 ± 2.8 | 68.9 ± 0.1 | 40.1 ± 0.5 | 63.6 ± 2.2 | 55.0 ± 2.5 | 48.3 ± 0.6 |
| GPT-2 Large | FlashAdamW | 40.1 ± 0.1 | 48.2 ± 0.7 | 22.8 ± 3.5 | 68.1 ± 0.1 | 40.8 ± 1.1 | 62.3 ± 0.4 | 56.3 ± 1.6 | 48.4 ± 0.8 |

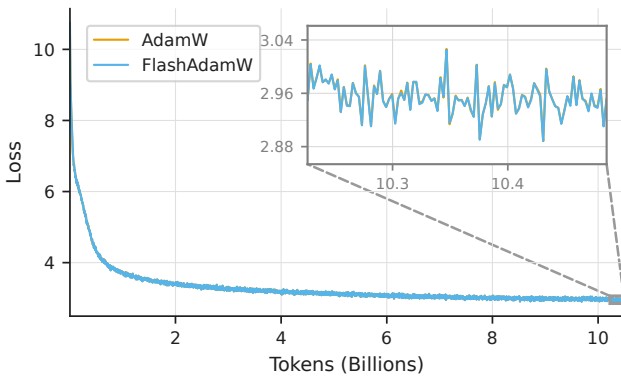

*(a)* GPT-2 Large, seed 18.

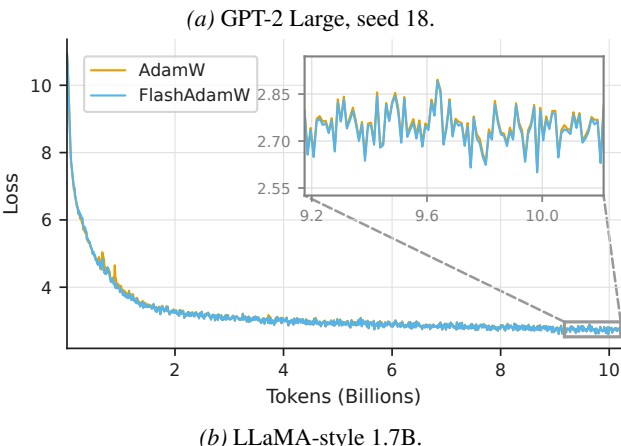

*(b)* LLaMA-style 1.7B.

*Figure 9.* **Additional LLM pretraining convergence.** AdamW and FlashAdamW training losses remain close for GPT-2 Large and the LLaMA-style 1.7B model, matching the behavior observed in the smaller GPT-2 124M experiments.

- **HellaSwag** (Zellers et al., 2019): A sentence completion benchmark requiring physical and temporal commonsense.
- **ARC-Easy** (Clark et al., 2018): The easy subset of the AI2 Reasoning Challenge, containing grade-school science questions.
- **CommonsenseQA** (Talmor et al., 2019): Multiple-choice questions requiring commonsense knowledge from ConceptNet.
- **PIQA** (Bisk et al., 2020): Physical Interaction Question Answering, testing physical commonsense reasoning.
- **OpenBookQA** (Mihaylov et al., 2018): Elementary science questions requiring multi-step reasoning over facts.

- **LAMBADA** (Paperno et al., 2016): Word prediction requiring broad discourse context understanding.
- **Winograd** (Levesque et al., 2012): Pronoun resolution problems requiring commonsense reasoning.
- **BoolQ** (Clark et al., 2019): Naturally occurring yes/no reading comprehension questions.

All benchmarks are evaluated in a zero-shot setting.

### B.4. LLM Finetuning

We fine-tune Llama-3.1-8B (Dubey et al., 2024) on OpenMathInstruct-2 (Toshniwal et al., 2024). For evaluation, we use GSM8k (Cobbe et al., 2021), a benchmark of 1,319 grade school math word problems that require multi-step arithmetic reasoning.

**Training and Evaluation.** We use FSDP2 (Feng et al., 2022) with full parameter sharding and activation checkpointing (Chen et al., 2016) applied to every transformer layer. We use the AdamW optimizer, with $\beta_1 = 0.9$ and $\beta_2 = 0.95$. The global norm is clipped at 1.0 and a weight decay of 0.1 is used. Weight decay is applied only to weight matrices, excluding biases, embeddings, and layer normalization parameters. We train for 5000 steps with an effective batch size of approximately 5.2 million tokens per step. We apply learning rate warmup for the first 1000 steps, followed by a cosine decay to 0.

We evaluate on the GSM8k test set using temperature $T = 0.2$ decoding. Following standard practice, we extract the final numerical answer from model generations and compare against ground truth.

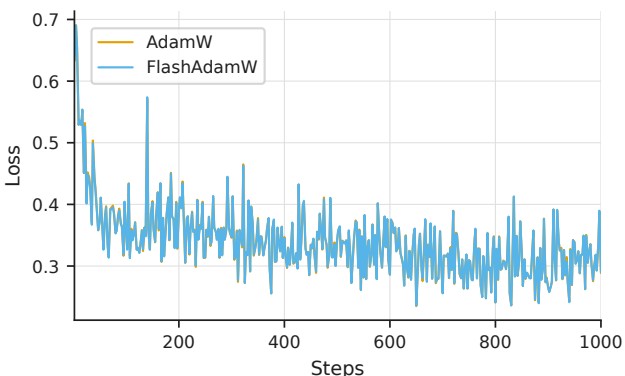

*Figure 10.* **Training convergence for LLM finetuning** (Llama-3.1-8B + AdamW). Comparison of training loss between reference AdamW and FlashAdamW during supervised finetuning on OpenMathInstruct-2.

