# OpenReview forum: "FlashOptim: Optimizers for Memory-Efficient Training"
_ICML.cc/2026/Conference — ICML 2026 spotlight_

### Official Review · Reviewer_LYjU · 2026-03-01

**Soundness:** 2
**Presentation:** 3
**Significance:** 2
**Originality:** 3
**Overall Recommendation:** 4
**Confidence:** 4

**Summary:**

This paper addresses the significant memory bottleneck caused by optimizer states and master weights in large-scale model training by introducing FlashOptim and Lion. To achieve this, the authors propose a Unit in the Last Place (ULP)-based weight splitting technique that reconstructs 32-bit master weights using a 16-bit downcast weight and an 8-bit or 16-bit integer error correction term, effectively minimizing reconstruction error by avoiding wasted exponent bits. Furthermore, the framework employs companded optimizer state quantization, which applies non-linear transformations—specifically a softsign-like function for momentum and a square root function for variance—to reshape heavy-tailed state distributions toward uniformity prior to standard 8-bit group-wise quantization. Empirical evaluations demonstrate that FlashOptim reduces the per-parameter memory footprint of AdamW from 16 bytes to 7 bytes (or 5 bytes with gradient release) without degrading convergence trajectories or downstream task performance across image classification (ResNet-50), language model pretraining (GPT-2), and supervised finetuning (Llama-3.1-8B).

**Compliance With Llm Reviewing Policy:**

Affirmed.

**Final Justification:**

The authors' rebuttal successfully addressed my primary concerns by providing critical baselines and convincing large-scale pretraining results, and while theoretical questions regarding long-term spectral dynamics and rank collapse remain, the robust empirical evidence justifies raising my score to a Weak Accept.

**Key Questions For Authors:**

1. Can you provide a head-to-head empirical comparison against established low-precision and sublinear memory optimizers, specifically 8-bit Adam, MicroAdam, or GaLore? Ideally, this should include metrics on peak memory savings, step-time throughput, and final validation loss on the pretraining tasks.

2. Is it possible to provide a formal convergence analysis for the proposed ULP-based quantized updates under standard non-convex optimization assumptions? Furthermore, have you investigated how the continuous injection of INT8/UINT8 truncation and companding noise affects the spectral properties of the weight matrices over extended training horizons?

3. Can you demonstrate the stability of FlashOptim on a pretraining task from scratch using a model in the 1B+ parameter regime (e.g., a 1.1B or 7B architecture)? The current 124M parameter pretraining experiment is insufficient to support the "Large-Scale" claim in the title, as quantization noise often behaves very differently at true scale.

**Limitations:**

yes

**Strengths And Weaknesses:**

Strengths:

1. The proposed memory reduction is highly practical and addresses a critical bottleneck in large-scale model training. Compressing the AdamW per-parameter memory footprint from 16 bytes down to 7 (or 5 with gradient release) without requiring complex system-level offloading is a highly valuable contribution to the community.

2. The design of the companding functions (e.g., $\phi_m(x) = \frac{2x}{1+|x|}$ and $\phi_v(x) = \sqrt{x}$) is remarkably elegant. Instead of relying on complex block-wise floating-point formats, the authors utilize a computationally cheap, one-line non-linear projection to uniformize heavy-tailed distributions prior to standard INT8 quantization. This simplicity makes it highly amenable to efficient Triton kernel fusion.


3. The empirical results are robust across a diverse set of tasks, including ResNet-50 training, GPT-2 pretraining, and Llama-3.1-8B supervised finetuning. The loss trajectories and downstream evaluation metrics (e.g., GSM8k) closely track the full-precision baselines, demonstrating the immediate viability of the method.



Weakness:


1. A flaw in the experimental design is the absence of direct comparisons against state-of-the-art memory-efficient optimizers. The authors establish their claims by contrasting FlashOptim solely with standard, full-precision SGD, AdamW, and Lion. To substantiate claims of practical superiority, it is mandatory to benchmark against low-precision and sublinear memory optimizers. Specifically, the paper cites but fails to empirically compare against 8-bit Adam[1], MicroAdam[2], GaLore[3], or recent orthogonalized update methods like Muon[4].

[1] Loshchilov, I. and Hutter, F. Decoupled weight decay regularization. arXiv preprint arXiv:1711.05101, 2017.

[2] Modoranu, I.-V., Safaryan, M., Malinovsky, G., Kurtić, E., Robert, T., Richtárik, P., and Alistarh, D. MicroAdam: Accurate adaptive optimization with low space overhead and provable convergence. Advances in Neural Information Processing Systems, 37:1-43, 2024.

[3] Zhao, J., Zhang, Z., Chen, B., Wang, Z., Anandkumar, A., and Tian, Y. GaLore: Memory-efficient llm training by gradient low-rank projection. In International Conference on Machine Learning (ICML), 2024a.

[4] Jordan, K., Jin, Y., Boza, V., You, J., Cesista, F., Newhouse, L., and Bernstein, J. Muon: An optimizer for hidden layers in neural networks, 2024.

2. A weakness of this submission is its entirely empirical nature, lacking any rigorous theoretical foundation or convergence analysis. While the authors introduce novel ULP-based weight splitting and non-linear companding functions, they provide zero mathematical proof that these quantized updates actually converge. In the current landscape of memory-efficient optimization literature (such as MicroAdam or GaLore, which provide clear convergence bounds), it is standard practice to mathematically demonstrate that the proposed approximation does not break the convergence guarantees of the base optimizers (e.g., SGD, AdamW) under standard non-convex optimization assumptions.


3. The title and introduction promote FlashOptim as a solution for "Large-Scale Training". However, the empirical validation for training from scratch is restricted to a small GPT-2 model with only 124M parameters , trained on a mere 10B tokens. By contemporary LLM standards, 124M parameters is a toy setting. It is widely documented in optimization literature that many low-precision or memory-compressed optimizers perform flawlessly at the 100M scale but suffer from severe training instabilities, irrecoverable loss spikes, or divergence when scaled to 1B+ parameters and trained over hundreds of billions of tokens.

---

> ### Author Rebuttal · Authors · 2026-03-31
>
> We thank the reviewer for the detailed critique.  We agree that comparisons to other compressed optimizers and larger scale experiments would strengthen the paper claims.
>
> **Missing baselines.** We have now run end-to-end comparisons against the two closest compressed baselines: Dettmers-2022 8-bit Adam and Zamirai-2020 / Warner-style BF16 splitting with Kahan correction.
>
> On finetuning, all methods are statistically equivalent:
>
> | Method | GSM8k acc. |
> |---|---:|
> | Reference AdamW | 75.09 ± 0.40 |
> | FlashOptim | 74.98 ± 0.77 |
> | Dettmers-2022 | 75.06 ± 0.20 |
> | Zamirai-2020 | 75.06 ± 0.30 |
>
> On GPT-2 pretraining, FlashOptim and Zamirai-2020 match the full-precision baseline, while Dettmers-2022 is slightly worse:
>
> | Method | Val loss | Mean ICL |
> |---|---:|---:|
> | Reference AdamW | 3.263 ± 0.001 | 44.0 ± 0.4 |
> | FlashOptim | 3.265 ± 0.001 | 45.0 ± 1.0 |
> | Zamirai-2020 | 3.264 ± 0.001 | 44.2 ± 0.4 |
> | Dettmers-2022 | 3.279 ± 0.000 | 43.0 ± 1.3 |
>
> The profiling picture is where the methods separate most clearly:
>
> | Variant | Param GiB | Optim GiB | Peak GiB | Step ms |
> |---|---:|---:|---:|---:|
> | Reference | 29.9 | 59.8 | 175.2 | 12.5 |
> | FlashOptim | 15.0 | 23.4 | 112.9 | 11.5 |
> | Zamirai-2020 | 15.0 | 44.9 | 134.3 | 15.8 |
> | Dettmers-2022 | 29.9 | 15.2 | 130.6 | 31.9 |
>
> The takeaway is that FlashOptim is the only method here that reduces both parameter memory and optimizer memory while preserving quality and step time.
>
> Methods such as MicroAdam, GaLore, and Muon are still relevant in the broader memory-efficient training landscape, and we will discuss that relationship more clearly in the revision. We did not compare to them because they change the optimization rule itself (e.g., low-rank projection, different preconditioning, or a different optimizer). FlashOptim's goal is narrower: preserve the update rule of standard SGD, AdamW, Lion, etc., while reducing the memory cost of their numerical representation. For that claim, Dettmers-2022 and Zamirai-2020 are the closest comparisons, and we will include them in the revised version.
>
> Hence, our goal is not to argue that a particular FlashOptim-enhanced optimizer is the best optimization algorithm. It is to argue that the FlashOptim enhancements are useful. One could construct FlashMicroAdam, FlashGalore, FlashMuon, etc, and might get even better results than what we obtain with Adam, SGD, or Lion.
>
> **Convergence analysis.** We agree that the current paper does not provide a full formal convergence proof for the fully quantized method, and relies on empirical experiments. Regarding convergence what we can currently support is the following:
> 1) **Algorithmic convergence excluding finite-precision effects:** FlashOptim does not alter the analytic update rule of the underlying optimizer; it changes only how weights and optimizer states are numerically represented. Therefore, standard convergence results for SGD, AdamW, Lion, remain the relevant reference point for the underlying update dynamics under their original assumptions.
> 2) **Master-weight representation error:** For the effect of using narrower master weights, we provide a closed-form characterization of when reconstruction error can become large enough to threaten the intended update in the limitations section.
> 3) **Companded state quantization:** We believe that SGD convergence is not affected by optimizer state quantization. For FlashSGD, if we assume the dequantization error is zero mean at every time step and has bounded second moment, then we can rewrite that dequantization error as an extra part of the usual stochastic gradient noise. With this reparameterization, FlashSGD reduces to the  SGD with momentum update with an additional gradient noise term from the dequantization error. Hence FlashSGD has the same convergence guarantee as standard SGD with momentum, except with a larger noise floor coming from an additional term of order $\mu^2\sigma_q^2$. We are not able to provide a detailed proof because of space constraints, but we will revise the appendix to include a more complete proof about the convergence of FlashSGD.
>
> **Pretraining scale.** We agree that the original from-scratch pretraining evidence was too small to justify a broad reading of "large-scale." In response, we ran an additional from-scratch pretraining experiment on GPT-2 Large (770M parameters) and Llama-1B (to include a more modern setting with RoPE and longer context lengths). In both scenarios we observed essentially identical convergence to the full-precision AdamW baseline and no meaningful degradation on our ICL evaluation.
>
> - [Figure for GPT-2 Large convergence](https://postimg.cc/2bdv0zvs)
> - [Figure for Llama-1B convergence](https://postimg.cc/mzWC3czb)
>
> We will add these experiments to the revision. We will also revise the abstract/introduction language to make the scope of the evidence clear.

---

> > ### Author Rebuttal · Reviewer_LYjU · 2026-04-02
> >
> > I thank the authors for their swift and highly constructive rebuttal. The new end-to-end baselines (Dettmers-2022, Zamirai-2020) properly contextualize the method's performance, and the impressive from-scratch pretraining results on GPT-2 Large and Llama-1B fully resolve my concerns regarding the paper's "large-scale" claims. Regarding the convergence analysis, while I appreciate the theoretical sketch, I still suspect that continuous quantization noise might interact with weight decay to accelerate rank collapse over extended training horizons; however, I concede that a spectral analysis is beyond the scope of this specific manuscript. Given the robust empirical validations and the authors' commitment to incorporating these new results and proofs into the revision, I am raising my score to a Weak Accept.

---

### Official Review · Reviewer_vXBy · 2026-03-06

**Soundness:** 4
**Presentation:** 4
**Significance:** 3
**Originality:** 3
**Overall Recommendation:** 5
**Confidence:** 5

**Summary:**

The authors improve the memory efficiency of standard optimizers (SGD with momentum, AdamW) with little impact on training step time, validation loss, and downstream task performance.

Key innovations are
1) a more careful technique for float splitting that removes redundant exponent bits
2) companding functions for transforming optimizer state before quantization.

These innovations have been implemented in such a way to fit with standard distributed training settings, and compared against fair baselines where common opportunities for performance optimization presented themselves.

The authors show that their memory efficient optimizers have no impact on training loss of a 124M GPT-2 model trained for 10B tokens, nor a Resnet-50 trained for 90 epochs on imagenet-1k, and no impact on downstream task performance on Imagenet-1k Top-1 Accuracy or maths reasoning on a finetuned 8B param Llama3 model. They also illustrate superior quantization error over weights and optimiser state over alternative methods.

**Compliance With Llm Reviewing Policy:**

Affirmed.

**Final Justification:**

As stated in my first review, I think this paper is essentially as good as it can be and is a clear accept. The authors cleared up a minor point about designing companding functions for optimizer state, but I don't think this is enough to raise score.

**Key Questions For Authors:**

I think the scope of the paper is too limited to justify raising my score further, and believe the paper is almost as good as it can be. A few minor points that would improve the paper:

1. Is there a principled way of choosing companding functions for optimizer state? For example [1] propose a cube-root density companding function for block scaling of weights as the optimal solution for weights modelled as student-t distribution with a fixed number of bits.
2. Do memory spikes occur in different places to standard training? For example, does reconstruction of master weights in the optimiser step create a memory spike, which offset the spike that would usually occurs by creating a downcast copy of master weights during a weight gather?
3. The experiment details say that dataset ordering was kept the same. Were weights also initialised to be the same?

[1] (Optimal formats for weight quantisation)[https://arxiv.org/abs/2505.12988]

**Limitations:**

yes

**Strengths And Weaknesses:**

**Strengths**
* Easy win for improving memory efficiency of optimisers with no impact on wall clock time or task performance
* Clear and well presented
* Fair comparisons with optimizations made to baselines
* Can help keep checkpoint storage costs down for large model training runs

**Weakness**
* Fairly unprincipled way of choosing companding function - unclear how you should choose companding functions for unknown optimisers
* Impact is restricted a bit to finetuning use cases (large models, small compute budgets) when unable to amortise memory across large numbers of accelerators
* Would have liked to see memory traces over the course of a training step, rather than just peak memory use.

---

> ### Author Rebuttal · Authors · 2026-03-31
>
> We thank the reviewer for the strong overall assessment and for the concrete questions.
>
> **Choosing companding functions.** We agree that this deserves a more principled explanation in the paper.
>
> In theory, the minimal-reconstruction-error quantization pipeline spaces the quantization bins in proportion to the input distribution raised to some power [1], where the power depends on your error metric. Because this is distribution-dependent, the best companding choice depends on the data, architecture, etc, and is not universal.
>
> In our current work, we therefore use a practical selection procedure rather than claiming a closed-form rule. We require a preprocessing function `f` satisfying:
> 1. f has a well-defined inverse, so that we can dequantize
> 2. Both f and its inverse must be cheap to evaluate within a fused optimizer kernel,
> 3. Applying f must reduce quantization error on representative tensors.
>
> Conditions (1) and (2) are easy to assess and substantially narrow the search space. To assess (3), we evaluated a shortlist of functions satisfying (1) and (2) on collected optimizer-state tensors from representative training runs across different domains.
>
> The recent "Optimal Formats for Weight Quantisation" result is interesting, but it addresses a different setting: designing quantization formats for model weights under a storage constraint. Our setting is optimizer-state quantization inside the training kernel, where the compander and its inverse run at every step. In this regime, kernel cost, exact invertibility, and ease of fusion matter just as much as quantization fidelity. We will clarify this distinction in the revision and state more explicitly that automatically deriving companders for new optimizers and tensor types is a promising direction for future work.
>
> **Memory traces over a training step.** FlashOptim does not introduce a new HBM memory spike during optimizer reconstruction. The fused kernels materialize FP32 master weights only inside the kernel and process parameters in blocks, so those reconstructed values are never written back to GPU HBM. In practice, the peak still occurs at the usual places in the training step: typically the start of the backward pass (no activation checkpointing), or end of the backward pass (with activation checkpointing). We will add this explanation and the full trace figure.
>
> This [GPU memory trace figure](https://postimg.cc/SJ2jgSW1) also makes the key qualitative point visible: FlashOptim lowers the constant parameter/optimizer footprint throughout training. We will update the manuscript to include the memory trace figure.
>
> **Initialization control.** Yes, weights were also initialized identically across runs. We agree this should have been stated explicitly, and we will add it to the experimental details.
>
> [1] Panter, P. F., & Dite, W. (1951). Quantization Distortion in Pulse-Count Modulation with Nonuniform Spacing of Levels. Proceedings of the IRE, 39, 44–48.

---

> > ### Author Rebuttal · Reviewer_vXBy · 2026-04-03
> >
> > Thanks for the additional detail on how to choose companding functions! It would be useful to see your shortlist, a quantization error table, and a bit of detail on how you generate representative tensors in an appendix.
> >
> > The authors have resolved my concerns.

---

### Official Review · Reviewer_QWYD · 2026-03-11

**Soundness:** 3
**Presentation:** 4
**Significance:** 4
**Originality:** 4
**Overall Recommendation:** 5
**Confidence:** 4

**Summary:**

The paper FlashOptim introduces a new method (based on two techniques: improved float splitting and companded optimizer state quantization, acommpanied by fused optimized kernels) to reduce the memory requirements in training. The authors use these techniques to reduce the memory requirements of frequently used optimizers like SGD (from 12 to 6 or 4 bytes) and Adam (from 16 to 7 or 5 bytes) s well as Lion, which helps to reduce the memory requirements during traingn.

The authors provide a comprehensive related work section that considers past and current methods in low-precision training, optimzer state compression, communication of gradients, memory efficient optimization methods, and system level memory optimizations. Here, they also discuss differences of FalshOptim to these approaches.

In the section weight spliiting, they discuss how they can reduce the memory consumption in mixed precision training by weight splitting, where they propose to neglect the exponent bits, which are not required diue to the properties of loating point numbers and rounding, which allows to scale and late on reconstruct values.

In companded optimizer state quantization, the authors propose to extend/improve groupwise quantization by using nonlinear companding functions before quantization that help to utilize quantization bits and reduce the quantization error.  They need to find such functions for momentum and variance tensors separately and provide algortihtms for these.

The implementation is performed using Triton and supports sharding for distributed training.

The authors perform a comprehensive evaluation, were they use their methods for training a resnet for image classification, perform LLM pretraining and finetuning.

In their analysis, they analyze the convergence  of the training, memory and speed. Here, the authors show that their method  introduce a neglectible error in classification results and LLM pretraining results, while achieving a significant reduction of the required memory usage. Furthermore, they analyze the reconstruction error of their approximation and show that the compading is required.

**Compliance With Llm Reviewing Policy:**

Affirmed.

**Final Justification:**

The rebuttal of the authors clarifed the remaining issues. Nevertheless, the do not increase the soundnessin such a way that an "excellent" rating is justified. Hence, the orignial ratings persist.I recomment to accept the paper.

**Key Questions For Authors:**

1. In which kinds of architectures do you expect might the quantization result in a degredation of performance?
2. How can the 24 bits master weights be exploited by GPUs? They typically support fixed data types (FP32, BF16, INT16, INT8..) hence, won't these just implicitly be resized to 32 bits?
3. On which kidn of task/architecture would you expect that your methods obtain worse performance than the "classical" algorithms?

**Limitations:**

Yes

**Strengths And Weaknesses:**

Strength:
The authors introduce two interesting methods that help to introduce memory usage for training significantly. They provide a solid theoretical discussion of their approaches, e.g. the underlying numerical propoerties and concrete algorithms.  These two ideas are quite intriguing, since they result in theoretically significant memory reductions.

This is also shown in the experimental evalaution. The authors obtain a significant reduction of memory usage in training for different tasks without sacrificing memory usage and speed (however, the comparison could be extended, as discussed below).

The athors provide an extensive overview about previous methods and provide clear distinctions of the differences of FlashOptim to these methods.

The evaluation is performed on three different tasks, on all these tasks, FlashOptim obtains interesting results.

Weaknesses:
The writing style of the paper is sometines to compressed, at certain points it is hard to follow the authors without diving into the provided references. Since the authors combine and extend different techniques, the underlying basiscs of their qork could be discussed in more detail. However, overall the paper is written well and follows a clear path.

The authors do not report timing results/speed for the image classification evaluation.

The comparison optimizers are reimplemented by the authors using Triton to ensure fairness. However, I think the authors should also compare the "classical" reference implementation to their implementation, because it is not possible to asess the "quality" of their implementation has an impact on performance.

The evaluation could be more comprehensive, i.e., at least two or more different NNs/LLMs should have been used for evaluation in each task (classification, LLM pretraining, LLM finetuning). Probably, the resuls would have been similar, but it is hard to assess if the reults are not just obtained by being "lucky" and the method performs worse on other NNs/LLMs. Even if this is unprobable, some more evidence is required here.

---

> ### Author Rebuttal · Authors · 2026-03-30
>
> We thank the reviewer for the careful reading and constructive suggestions. We agree that the manuscript can be clearer in a few places, and we will revise it accordingly. On timing for image classification specifically, these measurements are already reported in Table 6 of the Appendix, and we will point to them more clearly in the main text.
>
> **Reference implementation.** This is a good point. To verify that performance is not an artifact of our Triton baseline, we compared our 32-bit Triton AdamW directly against stock PyTorch AdamW with `fused=True`, which is the relevant optimized reference implementation. The step times are essentially identical: 10.2ms for our Triton AdamW versus 10.1ms for PyTorch AdamW. We will include this comparison explicitly in the revision.
>
> **Breadth of evaluation.** We agree that more architectures would further strengthen the experimental analysis. Our current evidence already spans three distinct settings: CNN training, language-model pretraining, and 8B supervised finetuning. In response to this review cycle, we are also adding compressed-baseline comparisons to further strengthen the analysis (see our reply to Reviewer CjkB), and additional LLM pretraining models at 1B scale (see our reply to Reviewer LYjU).
>
> **Limitations of quantization.** As outlined in our limitations section, if FlashOptim were to underperform relative to the classical optimizer, we would expect it first in numerically sensitive regimes where updates approach the quantization resolution: very late-stage training at tiny learning rates, extremely sparse-update parameters such as embedding layers, or optimizers whose state distributions differ substantially from the momentum/variance tensors we designed companders for. None of the empirical settings we evaluated showed degradation when training with FlashOptim.
>
> **Efficient 24bit values.** We do not rely on a native 24-bit hardware datatype. The forward pass, backward pass, and stored model parameters remain in standard BF16. During the optimizer step, we combine the BF16 parameter with the stored INT8 correction term inside the fused kernel to materialize a temporary FP32 update there, and then compress back to BF16 plus an updated INT8 correction term. Our storage scheme provides 24-bit weight semantics while still carrying out the optimizer update in FP32 completely within the kernel, without storing FP32 master weights in HBM. Outside the optimizer step the weights are stored as BF16 values, and the correction terms as INT8 tensors in the optimizer state.

---

> > ### Author Rebuttal · Reviewer_QWYD · 2026-04-02
> >
> > I thank the reviewers for their reply. I think this adresses my concerns.

---

### Official Review · Reviewer_CjkB · 2026-03-12

**Soundness:** 3
**Presentation:** 3
**Significance:** 3
**Originality:** 3
**Overall Recommendation:** 5
**Confidence:** 3

**Summary:**

The authors present FlashOptim, a suite of memory-efficient optimizers (supporting SGD, AdamW, and Lion) designed to significantly reduce the per-parameter memory footprint during neural network training. The proposed method reduces AdamW's memory requirement from 16 bytes to 5 bytes when combined with gradient release.

The paper introduces two core techniques:

- ULP-based Weight Splitting: Instead of storing a full 32-bit master weight, the method stores a 16-bit weight alongside an 8-bit or 16-bit integer error correction term. By exploiting the round-to-nearest property of floating-point arithmetic, the exponent of the error is inferred from the unit in the last place (ULP) of the downcast weight. This elegantly maximizes the utilization of the entire bit space within the correction term.

- Companded Optimizer State Quantization: The authors introduce lightweight, non-linear companding functions before standard 8-bit group-wise linear quantization. This approach effectively reduces the quantization error of the optimizer states.

**Compliance With Llm Reviewing Policy:**

Affirmed.

**Final Justification:**

The paper presents an interesting idea on a memory-efficient optimizer. All my concerns are carefully addressed.

**Key Questions For Authors:**

Please refer to the weaknesses.

**Limitations:**

Yes

**Strengths And Weaknesses:**

Strengths:

+ Interesting Weight Splitting Method: The paper introduces a very interesting approach to weight splitting by exploiting the Unit in the Last Place (ULP). Recognizing that the rounding error between the master weight and downcast weight is strictly bounded, the authors encode this error as a pure integer (INT8 or INT16) mapped to this narrow ULP interval, rather than using a wasteful floating-point format. This numerical representation reclaims wasted exponent bits and achieves high reconstruction accuracy with fewer total bits.

+ Effective Optimizer State Quantization: The proposed companded quantization method is both simple and highly effective. By applying computationally lightweight non-linear companding functions, the authors successfully reshape the notoriously heavy-tailed distributions of optimizer states into a more uniform spread. This effectively minimizes quantization error before applying standard 8-bit group-wise quantization, cleanly resolving the training divergence issues often seen in naive linear quantization.

+ Relatively Comprehensive Empirical Validation: The authors provide solid and diverse experimental evaluations across different domains. The method is validated through LLM pretraining on a 10B token dataset, supervised finetuning on Llama-3.1-8B, and vision classification training on ImageNet with ResNet-50. Furthermore, the paper thoroughly profiles both the memory footprint and the step latency. Thanks to the effective algorithmic compression techniques and the optimized fused Triton kernels, the proposed approach drastically reduces memory overhead without any degradation in model accuracy and without introducing any additional computational latency.

Weaknesses:

- Lack of End-to-End Comparison with Compressed Baselines: While the paper compares against uncompressed FP32 optimizers and provides component-level ablations, it crucially lacks end-to-end training comparisons against memory-efficient baselines mentioned in this paper: 1. BF16 Weight Splitting (e.g., Zamirai et al., 2020; Warner, 2024). 2. 8-bit Optimizers (e.g., 8-bit Adam by Dettmers et al., 2022). Providing end-to-end benchmarking (loss, metrics, memory, and throughput) against these state-of-the-art compressed counterparts is essential to validate FlashOptim as a superior drop-in replacement.

---

> ### Author Rebuttal · Authors · 2026-03-30
>
> We thank the reviewer for the positive assessment. We agree that comparisons to other compressed optimizers strengthen the paper.
>
> **Compressed baselines.** We have now run the two closest baselines the reviewer asked for:
> - Dettmers-2022 (aka 8-bit Adam): compresses optimizer state to INT8, but keeps FP32 master weights.
> - Zamirai-2020 (BF16 splitting + Kahan correction): compresses master weights, and stores a correction term.
>
>
>
> **End-to-end quality.** On supervised finetuning, all methods are within run-to-run variance:
>
> | Method | GSM8k acc. |
> |---|---:|
> | Reference AdamW | 75.09 ± 0.40 |
> | FlashOptim | 74.98 ± 0.77 |
> | Dettmers-2022 | 75.06 ± 0.20 |
> | Zamirai-2020 | 75.06 ± 0.30 |
>
> On GPT-2 pretraining, FlashOptim, Zamirai-2020, and full-precision AdamW remain tightly clustered, while Dettmers-2022 is slightly worse:
>
> | Method | Val loss | Mean ICL score |
> |---|---:|---:|
> | Reference AdamW | 3.263 ± 0.001 | 44.0 ± 0.4 |
> | FlashOptim | 3.265 ± 0.001 | 45.0 ± 1.0 |
> | Zamirai-2020 | 3.264 ± 0.001 | 44.2 ± 0.4 |
> | Dettmers-2022 | 3.279 ± 0.000 | 43.0 ± 1.3 |
>
>
>
> **Memory and throughput.**
>
> | Method | Param GiB | Optim GiB | Peak GiB | Step time (ms) |
> |---|---:|---:|---:|---:|
> | Reference AdamW | 29.9 | 59.8 | 175.2 | 12.5 |
> | FlashOptim | 15.0 | 23.4 | 112.9 | 11.5 |
> | Zamirai-2020 | 15.0 | 44.9 | 134.3 | 15.8 |
> | Dettmers-2022 | 29.9 | 15.2 | 130.6 | 31.9 |
>
> FlashOptim is the only method here that reduces both parameter and optimizer memory while maintaining comparable step time to the reference optimizer. We will add these end-to-end comparisons and profiling results to the revision.

---

> > ### Author Rebuttal · Reviewer_CjkB · 2026-04-02
> >
> > The authors have resolved all my concerns.

---

### Decision · Program_Chairs · 2026-04-30

**Decision:**

Accept (spotlight)

**Comment:**

### **Summary of Contributions**
The paper introduces FlashOptim, a suite of memory-efficient optimizers (including SGD, AdamW, and Lion) designed to significantly reduce memory requirements during neural network training. The method reduces the per-parameter memory footprint of AdamW from 16 bytes to 7 bytes, or 5 bytes with gradient release. This is achieved via two primary techniques: ULP-based weight splitting, which pairs a 16-bit downcast weight with an 8-bit or 16-bit integer error correction term, and companded optimizer state quantization, which applies non-linear companding functions to reshape heavy-tailed optimizer states prior to 8-bit quantization.

### **Decision Reasoning**
The submission received a strong positive consensus from the reviewers (Accept, Accept, Accept, Weak Accept).

* **Strengths:** Reviewers agreed that the ULP-based weight splitting and companding functions are mathematically elegant and highly effective. The use of optimized fused Triton kernels ensures that the memory footprint is drastically reduced without degrading model accuracy or introducing step latency.
* **Rebuttal Impact:** The authors successfully addressed all major concerns raised during the review phase:
    * **Baselines:** In response to requests for established baselines, the authors provided end-to-end comparisons against 8-bit Adam (Dettmers) and BF16 splitting (Zamirai). Results demonstrated that FlashOptim is the only method to reduce both parameter and optimizer memory while preserving step time and quality.
    * **Evaluation Scale:** To address concerns regarding the limited scale of the initial experiments, the authors ran an additional from-scratch pretraining experiment on a Llama-1B architecture.
    * **Memory Spikes:** The authors clarified that HBM memory spikes are avoided because FP32 master weights are materialized only inside the kernel and processed in blocks.
* **Limitations and Future Work:**
    * **Theoretical Formalism:** While empirical results are robust, a complete formal convergence analysis under standard non-convex assumptions remains an open problem, though a theoretical sketch for FlashSGD was provided during the rebuttal.
    * **Scope of Optimizers:** The current FlashOptim techniques are demonstrated exclusively on element-wise optimizers. It remains an open question whether these quantization and weight-splitting mechanisms can be effectively adapted for matrix-based optimizers, such as Muon. Investigating this extension serves as a clear direction for future work.

### **Conclusion**
The paper presents a technically solid and highly practical solution to a critical bottleneck in large-scale model training. The authors' rebuttal comprehensively resolved reviewer concerns regarding baselines and empirical scale. The submission is a clear accept, with the extension of the methodology to matrix-based optimization offering a promising avenue for subsequent research.